# Race, Ethnicity, and Geography as Determinants of Excessive Weight and Low Physical Activity in Pediatric Population: Protocol for Systematic Review and Meta-Analysis

**DOI:** 10.3390/healthcare12181830

**Published:** 2024-09-13

**Authors:** Yauhen Statsenko, Darya Smetanina, Gillian Lylian Simiyu, Maroua Belghali, Nadirah Ghenimi, Guido Hein Huib Mannaerts, Leena Almaramah, Maryam Alhashmi, Nazia Chun Mohammad, Rahaf Al Hamed, Sara F. Alblooshi, Khawla Talbi, Maitha Albreiki, Fatima Alkaabi, Anna Ponomareva, Milos Ljubisavljevic

**Affiliations:** 1Imaging Platform, ASPIRE Precision Medicine Institute in Abu Dhabi, United Arab Emirates University, Al Ain P.O. Box 15551, United Arab Emirates; gl.simiyu@uaeu.ac.ae; 2Department of Radiology, College of Medicine and Health Sciences, United Arab Emirates University, Al Ain P.O. Box 15551, United Arab Emirates; 202111407@uaeu.ac.ae (L.A.); 202201010@uaeu.ac.ae (M.A.); 700039035@uaeu.ac.ae (N.C.M.); 201802250@uaeu.ac.ae (R.A.H.); 202108214@uaeu.ac.ae (S.F.A.); 202050894@uaeu.ac.ae (K.T.); 202014271@uaeu.ac.ae (M.A.); 202105129@uaeu.ac.ae (F.A.); 3CIAMS Laboratory, Orléans University, 45062 Orléans, France; marouabelghali@yahoo.fr; 4Department of Family Medicine, College of Medicine and Health Sciences, United Arab Emirates University, Al Ain P.O. Box 15551, United Arab Emirates; nghenimi@uaeu.ac.ae; 5Mediclinic Al Ain Hospital, Khalifa Street, Al Ain P.O. Box 14444, United Arab Emirates; dr.mannaerts@gmail.com; 6Scientific-Research Institute of Medicine and Dentistry, Moscow State University of Medicine and Dentistry, Moscow 127473, Russia; lara12346@yandex.ru; 7Department of Physiology, College of Medicine and Health Sciences, United Arab Emirates University, Al Ain P.O. Box 15551, United Arab Emirates; milos@uaeu.ac.ae; 8Neuroscience Platform, ASPIRE Precision Medicine Research Institute Abu Dhabi, Al Ain P.O. Box 15551, United Arab Emirates

**Keywords:** pediatric obesity, physical activity, sedentary lifestyle, geography, race, ethnicity

## Abstract

The rationale for the current study is the sparsity of data on the combined effect of the environmental and individual risks of obesity and sedentary lifestyle in children of different races/ethnicities from different regions. An effective weight management strategy is hard to design due to insufficient evidence. This work was initiated to study race, ethnicity, and geography as determinants of excessive weight and low physical activity in the pediatric population. To achieve this aim, we systematically review publications on daily length of physical activity of light, moderate, and vigorous intensity, as well as sedentary time and BMI and its dynamics in children of different races/ethnicities and geographies. The extracted data are stratified into six major geographic regions and six races/ethnicities. Then, a random-effects meta-analysis is used to calculate the pooled mean of each outcome measure. A ridge regression is constructed to explore age-related change in BMI. A Kruskal–Wallis H test is applied to compare the pooled duration of physical activity and sedentary time in the subgroups. Finally, we calculate paired correlation coefficients between BMI and physical activity/inactivity for each group. The findings can be further used in public health surveillance to clarify the epidemiology of obesity, to guide priority setting and planning, and to develop and evaluate public health policy and strategy.

## 1. Introduction

According to the World Health Organization (WHO), the global prevalence of obesity has tripled within the last four decades, and the situation is expected to deteriorate because of the alarming rates of childhood obesity. In 2022, the proportion of children who were overweight exceeded 20%. This included 37 million youngsters under the age of 5 and 390 million minors aged from 5 to 19 years [1].

We intend to study the current obesity status and physical fitness levels in pediatric communities. The prevalence of obesity varies across geographic regions. In the previous two decades, the epidemic reached a plateau in Western countries and shifted to Asia and sub-Saharan Africa [2]. Epidemiology statistics constantly change; the maximum decennial increase in body mass index (BMI) was reported for the Oceania region. Between the 1970s and 2010s, the decennial increase in BMI was around 0.38 in South Asia and Central, East, and West Africa; 0.64 in Polynesia; 0.95 in Central Latin America; 1.00 in Eastern Europe; and 1.25 kg/m^2^ in Micronesia [3]. Between 1985 and 2019, the decennial increase in BMI was 0.15 in Japan, Russia, and Denmark and 0.88 kg/m^2^ in Malaysia and Oceania [4]. Still, the data for different regions and periods are not systematized.

Studies on the determinants of obesity have been inconclusive. Change in physical activity (PA) may underlie a shift in the epidemic landscape [5]. Only 19.7% of children perform at least 60 min of moderate-to-vigorous PA daily, which is a general WHO recommendation for their age [6,7,8]. The suggested activity may comprise brisk walking, running, cycling, and playing sports [9]. Adherence to WHO guidelines was the lowest in China and the highest in Finland (15.9% and 53%, respectively). It is recommended that adolescents undertake 2 h of overall PA a day, which is not followed in many countries [10,11,12]. In Europe, 14-year-old children are inactive for over 400 min/day [11]. In China, an active lifestyle is the choice of 9.5% of children, and the rest of the pediatric population displays sedentary behavior (SB) [13]. However, existing studies provide disparate findings, and the measurement of baseline activity is not unified. Hence, it is challenging to establish who is sedentary.

Industrialization is an environmental determinant of obesity; PA decreases with the availability of infrastructure facilities [14]. As a result, children treat an active lifestyle as an option rather than a necessity [5]. Higher levels of PA are recorded among those who enjoy active games with peers or participate in professional sports. However, the majority of youngsters prefer nonactive computer games. International migration is another factor modifying individual attitudes toward health-related behavior, as children are highly susceptible to environmental and societal changes [15].

Race/ethnicity is a bio-psycho-social determinant closely linked with the level of PA. This determinant is commonly studied in countries with a high influx of migrant populations. Still, the results are sparse and disparate. In the UK, preschool children of White and Asian races did not differ in the amount of sedentary time [16]. In the US, Hispanic children spend more time watching TV than their peers of other ethnic origins [17]. The same tendency for the Hispanic cohort is observed in the 6–11-year and 12–15-year age groups. Meanwhile, in the age range of 16–19 years, Blacks are the least active adolescents [18]. Levels of race/ethnic diversity differ among countries, which complicates this kind of research.

Few studies on obesity and PA consider environmental and individual risks, which limits the practical implications of the results. Recent meta-analyses were restricted to a single confounder and did not cover a combination of risks [19,20]. Moreover, many systematic reviews do not consider race, ethnicity, and geography as risk factors for obesity [21,22]. Publications focus on the following determinants of excessive weight: built environment, individual beliefs and attitudes towards a healthy lifestyle, interpersonal risks, institutional norms, and the role of communities [23]. Experts also highlight the necessity of considering individual genetic profile when researching the causes and prevention of obesity [24]. Multiple genetic risk factors for obesity are race-specific [25]. At the same time, geography defines socio-economic status, the production and availability of food, and access to sport facilities, which also contribute to health outcomes. Genetics and the environment do not act independently; therefore, it is necessary to study the combined effect race/ethnicity and geography on the development of adiposity. A fundamental systematic review is required to tackle obesity by designing an effective weight management strategy and developing a policy to curb it. The proposed meta-analysis outlines trends in PA and BMI of children in different regions and ethnic groups.

## 2. Objectives

The primary objective of this systematic review is to study race/ethnicity and geography as determinants of excessive weight and low PA in the pediatric population.

To address the objective, we designed the following tasks:Model the age-related dynamics of BMI in the pediatric population of different races/ethnicities and geographies.Study race/ethnicity and geographic location as determinants of weight and PA in children and adolescents.Look for associations between weight and PA in different races/ethnicities and geographies.

## 3. Methods and Analysis

To prepare the manuscript, we followed the Preferred Reporting Items for Systematic Review and Meta-Analyses Protocol (PRISMA-P) checklist (see Appendix A [26]). The study is filed in the International Database of Prospectively Registered Systematic Reviews (PROSPERO) with registration number CRD42024557213.

### 3.1. Study Design and Data Source

The project will review the literature on daily PA, sedentary time, and BMI and its dynamics in the pediatric population of different races/ethnicities and geographic regions. For the systematic search, we will use the PubMed search engine and the Web of Science, Scopus, EMBASE, SciELO, and LILACS biomedical databases. The query will comprise the following keywords: “BMI, physical activity, children, pediatric population, continent, country, ethnicity, race, sedentary time, exercise”. The search terms will be adapted for each selected biomedical database. We will retrieve papers published in any language from January 2000 to March 2024. We decided to investigate articles published during this time frame because of the dramatic impact of economic transformations on the population’s lifestyle in the late 20th century [27]. This change induced a rapid increase in rates of obesity among children and adolescents. Subsequently, the International Obesity Task Force recommended categorizing children and adolescents as overweight or obese according to their BMI. In 2000, Cole et al. proposed new cut-off values for BMI to define a child’s weight status [28].

Methods to assess adiposity include the use of z-scores, BMI-to-age percentiles, and raw BMI scores. All of them have merits and limitations. The use of scores is a method of choice in research due to its high precision. In clinical practice, BMI percentiles are predominantly used because they indicate a growing child’s BMI relative to other children of the same sex and age. We selected raw BMI scores as the primary outcome measure because many studies have reported it to reveal the effect of interventions in childhood obesity [29]. We will treat BMI values as a continuous variable without applying a threshold for different weight status categories.

### 3.2. Eligibility Criteria

Our review will target original peer-reviewed articles that examine BMI and PA in 5 to 17-year-olds. The sources should report the age of participants, absolute values of BMI, amount and type of PA. The selected articles must include data on the study location, race, and/or ethnicity. The participants should be free from primary metabolic syndrome, endocrine disorders, major depressive disorder, physical and mental disability. We will not consider reviews, meta-analyses, preprints, editorial letters, dissertations, case studies, or case series.

### 3.3. Study Records

Selection process: Covidence software will be used throughout the selection process. Articles matching the search query will be uploaded to the software for automatic deduplication. Then, two reviewers will perform parallel screening of titles and abstracts of the remaining papers. A third reviewer will affirm the final decision if the reviewers disagree on whether an article should be included in the study. Full-text screening will be conducted to compile a final list of papers eligible for the systematic review. We will hand screen the references in the selected publications to find articles not captured by the search strategy. The hand-screened studies will be uploaded to Covidence software to ensure the correct presentation of the selection process in a PRISMA flow chart. The reasons for excluding studies will also be documented and specified in the flow chart.

Data extraction: Two reviewers will use an online template to enter study characteristics and targeted variables. Extracted information will include the authors’ names; year of publication; country; study design; sample size; participant races/ethnicities as well as ages; and methods of data collection (e.g., self-reported surveys or wearable devices). Outcome measures will cover children’s BMI, duration of sedentary time, and PA duration in combination with its intensity based on the Metabolic Equivalent of Task (MET) [30]. The duration will be converted to minutes per day. Once the template is filled in, the third reviewer will verify 25% of the information for quality control.

Quality assessment of individual studies: Two reviewers will use a Joanna Briggs Institute checklist for cross-sectional and cohort studies to assess the quality of the papers included in this review. If any disagreement arises, the third reviewer will resolve it.

Dealing with publication bias: To deal with publication bias and compare studies with different methodologies, we will use graphical and statistical approaches. Begg’s and Egger’s tests will be applied to construct funnel plots for visual assessment of reporting bias [31,32]. An asymmetric distribution of effect sizes indicates between-study heterogeneity. Then, we will use the “trim and fill” method to identify the number of studies needed to construct a symmetric funnel plot [33].

### 3.4. Data Analysis and Synthesis

We will follow the United Nations classification of countries and territories to group study locations into six major geographic regions (Africa, Asia, Europe, Latin America and the Caribbean, North America, and Oceania) [34]. The National Institutes of Health’s definition of race and ethnicity will be used to classify datasets according to the following six biological origins: American Indian or Alaska Native, Asian, Black/African American, Hispanic/Latino, Native Hawaiian/Other Pacific Islander, and White [35].

We expect to receive data of a fragmented nature and varying quality. The between-study heterogeneity will be assessed with the Higgins–Thompson $I^{2}$ test [36]. $I^{2}$ over 75% indicates a high level of variability between the results of individual studies. We will perform a subgroup analysis to minimize the effect of study settings on the results. The studies will be divided according to geography and race/ethnicity, and the data will be analyzed separately for each subgroup.

To address the first specific objective, we will perform a pooled analysis of BMI in children and adolescents of different ages, sexes, races/ethnicities, and geography. Then, we will build a ridge regression model with a linear least squares optimization function. This approach was selected because it addresses the problem of multicollinearity—a situation where the variables are linearly dependent [37]. The model will allow us to study age-related dynamics in BMI. We will also examine sex differences in the growth trajectories of BMI.

Working on the second specific objective, we will explore the duration and intensity of PA in the pediatric population. The results will be presented as absolute mean values. We will also compute the percentage of the total estimated duration of PA and sedentary time. If a single study reports the level of PA for different subcohorts, the data will be clustered with the random-effects model. We resorted to this method because the true effect can vary from study to study due to the mixes of participants and assessment protocols [38]. The same model will be applied to conduct the final meta-analysis for different subgroups (regions and races/ethnicities). Then, we will employ a Kruskal–Wallis H test to compare the pooled values of sedentary time and moderate and vigorous PA in the pediatric populations of different continents and races/ethnicities. For each racial/ethnic group and geographic region, BMI values will be similarly analyzed.

For the third specific task, we will correlate the findings on BMI and PA for different races/ethnicities and geographies. To examine the association, we will calculate paired coefficients of correlation and construct a regression plot.

### 3.5. Methodology of Pilot Review

We followed a general recommendation to pilot a systematic review with a small sample of papers [39]. The review was carried out fully from data extraction to evidence synthesis. To this end, we collected relevant data from recent studies on pediatric obesity. Using the Google Scholar search engine, we obtained a list of 300 papers matching the following keywords: “BMI, physical activity, children, pediatric population”. Application of the inclusion and exclusion criteria yielded 263 observations from 100 articles for data extraction and analysis [6,10,11,16,17,18,40,41,42,43,44,45,46,47,48,49,50,51,52,53,54,55,56,57,58,59,60,61,62,63,64,65,66,67,68,69,70,71,72,73,74,75,76,77,78,79,80,81,82,83,84,85,86,87,88,89,90,91,92,93,94,95,96,97,98,99,100,101,102,103,104,105,106,107,108,109,110,111,112,113,114,115,116,117,118,119,120,121,122,123,124,125,126,127,128,129]. The characteristics of the articles are reported in Appendix A.

Among the pilot sample, the majority of the studies were conducted in North America [53,68,83,130] and Europe [40,42,54,56]. Ethnic/racial groups predominately included Hispanics, non-Hispanic Whites, non-Hispanic Blacks, Blacks, and Asians. Many studies investigated SB of children [6,120,121,122,123], and fewer papers analyzed moderate PA (MPA) [79,129,131]. A few publications explored the duration of vigorous PA (VPA) [50,125].

The reasons for methodological heterogeneity among the studies are outlined as follows. First, sex differences in BMI and PA were reported in a non-uniform manner. Some authors published data for entire study cohorts [47,50,54] whereas others specified findings for girls and boys [41,46,55]. Second, study methodologies and data collection varied widely. For example, devices for monitoring PA were either worn permanently or removed during swimming, bathing, or sleeping [40,73,109,120,131]. Several publications did not report activities at school [16,46]. Due to the heterogeneity of methodologies, the undocumented time a day ranged from 59.7% in South America to 77% in North America.

The pilot review was conducted to test whether the collected secondary data can reveal a significant difference in BMI and levels of PA among children of different races/ethnicities and in different world regions. To this end, we calculated mean BMI, duration of PA, sedentary time, and their standard deviations with a simple mathematical formula incorporated in Python [132]. However, this approach does not consider inter-study variability. Therefore, a random-effects model will be utilized in the full-scale meta-analysis.

## 4. Results of Pilot Review

### 4.1. Age- and Sex-Related Differences in Growth Trajectories of BMI

The regression analysis showed linear growth in BMI with advancing age. The mean BMI value was around 15 kg/m^2^ in 5-year-olds, and it rose to nearly 24 kg/m^2^ at 18–19 years. According to the model, BMI was close to the threshold of overweight (25 kg/m^2^) in late adolescents. In both sexes, the association between age and BMI was positive (r = 0.6718; *p* < 0.05), but it was tighter in boys (see Figure 1). The findings reflected trends in original longitudinal studies that indicate an increase in BMI with age [133]. This dynamics could be associated with a lack of nutrition literacy in children. Consequently, the pediatric population chooses calorie-dense foods and prefers sedentary activities in their free time.

### 4.2. Regional and Ethnic Disparities in BMI

We noted a high variability in pediatric BMI among different regions (see Figure 2). The lowest BMI was in children in Africa (17.12 ± 1.35 kg/m^2^), and the highest was in North America (20.49 ± 2.80 kg/m^2^). The range of BMI values was narrower in South America and wider in the Middle East, North America, and Europe. The distribution of BMI values was almost symmetric in most continents, but it was skewed toward higher values in South America. The data on different continents were reported disproportionally. We found 70 papers that covered information on North America, 67 on Europe, 27 on the Middle East, 17 on Asia, 13 on Australia/Oceania, 11 on Africa, and 8 on South America. The lowest BMI values were observed in Europe and the Middle East (15.8 kg/m^2^), and the maximal level was also reported in the Middle East (34.6 kg/m^2^).

BMI differed significantly among races/ethnicities (Figure 3). Hispanic children had the greatest BMI (22.04 ± 3.29 kg/m^2^), while Asians showed the lowest index value (18.64 ± 1.92 kg/m^2^). Blacks had the largest variability in BMI compared among all races, although the number of articles that reported findings on Blacks was smaller than those reporting on Whites (50 vs. 16, respectively). Hence, the number of articles did not influence the final result, and we obtained reliable information on the dispersion of data across races/ethnicities.

Racial/ethnic and geographic variability in BMI has been confirmed by many authors. Disparities in weight status are associated with exposure to psycho-social and behavioral risk factors for obesity, differences in environment, lifestyle, genetic ancestry, and other factors [134,135]. Education and access to healthcare also differ significantly, which shapes individuals’ knowledge, perception about healthy eating, and necessity of exercising. Moreover, socio-economic status can influence the relationship between BMI and race/ethnicity and region [136]. These findings confirm the complexity of research on obesity.

### 4.3. Physical Activity of Children in Different Regions

The preliminary analysis revealed a high prevalence of sedentary time over the time spent in moderate or vigorous activity (378.12 ± 167.07 vs. 52.55 ± 0.64 and 11.85 ± 7.93 min/day, respectively). The longest duration of sedentary time was in Asia, and the shortest was in North America (511.33 ± 61.30 vs. 262.22 ± 134.89 min/day, respectively; see Figure 4). Children in Africa spent more time in MPA and VPA than the rest of the pediatric population. The lowest level of MPA was found among children from Asia, whereas the minimum VPA was recorded in the European cohort (34.06 ± 15.72 and 11.17 ± 7.92 min/day, respectively).

Children from Asia were sedentary for 35.5% of the day, and this was the largest percentage in our research. They also spent the least time in MPA (2.4%), which is almost twice as low as in African children (4.6%). The involvement of children in Asia in VPA was also low—0.8%. Meanwhile, the shortest duration of vigorous activity was typical for North America (0.6%) and the longest for Africa (1.4%) (see Figure 5).

### 4.4. Racial/Ethnic Differences in Physical Activity

Among four racial/ethnic groups, Black children were the least active (see Figure 6). Asians, Hispanics, and Whites did not exhibit a pronounced difference in sedentary time (see Table 1). In all racial/ethnic groups, the amount of MPA and VPA was low compared to sedentary time. This is evident from the absolute number (see Table 1) and the shares of time for different activities (see Figure 6).

### 4.5. Association of Sedentary Time and Physical Activity with BMI

BMI and age are positively correlated due to the general trend towards a rise in the index with advancing age (r = 0.67; *p* < 0.01; see Figure 7 and Figure 8). The daily duration of sedentary time and exercising did not change with age for any ethnicity.

It is challenging to interpret the findings on the association of BMI with the duration of activity or inactivity. For example, the longest sedentary time was seen in the pediatric population of Asia or in children who were Asian by race/ethnicity. However, BMI was the lowest in these groups. Although Hispanic children were most active, their BMI value was the highest in our study.

Physiological and biochemical outcomes of PA could explain the tendencies in body-mass parameters. We observed a marked negative correlation between BMI and VPA (r = −0.52, *p* = 0.0009), VPA-to-sedentary time proportion (r = −0.51; *p* = 0.0061), and the proportion of total PA to sedentary time (r = −0.50; *p* = 0.0069). Long-lasting PA stimulates lipolysis; this is mainly true for light-intensity training. Hence, the lighter the endurance training is, the lower the weight is expected to be and vice-versa. However, most studies do not report findings on the amount of light-intensity training in the population.

We found a positive correlation between the ratio of moderate-to-vigorous PA and BMI (r = 0.47; *p* < 0.01). Because of the marked age-related change in BMI, we also focused on the dynamics of PA while a child is growing. The age-related dynamics in PA were also an issue in the current study. Age exhibited a pronounced inverse correlation with the proportions of vigorous and total PA with sedentary time (r = −0.57; *p* = 0.0002 and r = −0.53 *p* = 0.0006, respectively; see Figure 7 and Figure 8). In general, children become less active with age, and, reasonably, their BMI rises.

## 5. Discussion

### 5.1. Age-Related Dynamics of BMI

In the pilot study, we focused on BMI as a widely used tool to identify individuals at risk of obesity. However, authors do not agree on whether BMI can be transparently applied to children because of its substantial change during the growth period [137]. Children that have an equal BMI may differ in body composition, which depends on age, race, and sex [138,139]. The proposed meta-analysis will also include data on body composition. Different techniques are used to evaluate a person’s weight by breaking it down into core components. However, some of them are not common in clinical practice [140] or research due to intra- and inter-observer variability [141]. Therefore, we will limit the search to the findings of bioimpedance analyses.

Several statistical approaches are used to overcome difficulties in assessing a child’s weight status with BMI. Obesity can be monitored with raw BMI units, z-scores, and percentiles [142]. For children below 5 years of age, the WHO suggests using standardized age-for-sex BMI z-scores, which describe if a case falls within the standard deviation of the mean [143]. Percentiles are used to compare child measures to a reference population [144]. Although percentiles are easier to use in clinical practice, they may misclassify children’s weight status. BMI z-scores are a better tool for the assessment of adiposity in a single-time measurement; however, these scores are not recommended for longitudinal assessment. Raw BMI adjusted for age is an optimal solution for monitoring change in adiposity over time [142].

For the monitoring of pediatric obesity, researchers also compute the change in BMI. An approximation of BMI change can be obtained with LOWESS smoothing regression models [145] and latent-class growth analysis [146]. Regression models reflect the overall trend in body composition, whereas the other two methods show critical time points at which BMI reaches a plateau or rapidly increases. LOWESS smoothing is suitable for indicating adiposity rebound—the age at which BMI drops to its minimum in early childhood and starts rising again. Latent growth analysis plots BMI against the age-group axis and depicts complex growth patterns for the entire population and its subgroups [147]. The latter two methods do not work well with small samples [148]; therefore, we selected the ridge regression model to explore BMI-to-age associations.

### 5.2. Approaches to Study Geographic and Ethnic Disparities in BMI and Physical Activity

Original studies on region- and ethnicity-specific differences in BMI and PA fall into three groups. The first group of publications reports data from several independent research centers from different states. For example, Katzmarzyk et al. collected information about BMI and PA from 12 countries [50]. The second group provides findings on a diverse population of polyethnic countries, such as the US, the Russian Federation, and Australia [47,52,149,150]. These publications stratify study participants according to their race/ethnicity [17,55] and nativity/immigrant status [54,149]. The remaining group reports data from predominantly monoethnic countries, such as Spain and Poland. To obtain global statistics, we will combine findings from studies in all three groups in a meta-analysis.

Multicenter and multinational articles may shed light on disparities in PA and BMI among children. The findings of individual articles cannot be generalized beyond the studied population. Meta-analysis is the optimal solution to address the question of whether race/ethnicity and geography affect the level of PA and BMI of children. With this research methodology, we can synthesize the results using data pooled from original papers.

### 5.3. Methods to Examine Level of Physical Activity

The WHO classifies the intensity of PA into the following three types: light, moderate, and vigorous. Most papers do not report the full set of parameters that we target, namely inactive time and PA of light, moderate, and vigorous intensity according to MET [151]. For example, Hodgkin et al. and Gordon et al. examined the daily duration of sedentary time [17,51,52]. Laguna et al. reported findings on MPA and VPA but did not provide data on SB [56]. With our systematic review, we will extract all the available information, since ignoring any parameter can skew the analytics.

The duration and intensity of PA are commonly studied with accelerometers or pedometers. These devices precisely record some parameters of PA, but they do not provide contextual information about the type of exercise or recreation. The methodology of cohort studies varies considerably in terms of the period monitored with an accelerometer/pedometer. Participants are asked to wear devices during either waking hours or for the entire day. No consensus exists on the duration records that is sufficient to reflect a day of an individual [152]. This accounts for methodological heterogeneity among studies.

Self-administered questionnaires are another popular way of measuring PA. Quantitative surveys are time- and cost-efficient compared to wearable devices. The Global Physical Activity Questionnaire and International Physical Activity Questionnaire are the most common instruments used to assess PA level [153]. The surveys ask participants to recall the intensity and frequency of PA during a typical week. Scientists have also developed an ad hoc survey to explore contextual factors of PA. The results can be further used in public health surveillance to monitor and clarify the epidemiology of obesity. They will help guide priority setting and planning to develop public health policy and strategy [154].

### 5.4. Role of PA in Tackling Obesity

Our pilot study indicated that Asian children had the lowest BMI despite the smallest amount of PA performed. The results imply that the effect of PA on weight differs among individuals. Personalized weight-management interventions necessitate the identification of parameters of PA required to initiate weight loss. Previous studies proved the effectiveness of resistance exercise and combined aerobic–resistance training in the management of adolescence obesity [155,156]. After 6 months, participants experienced improvement in blood pressure and decreases in BMI and pulse-wave velocity [155]. Regular resistance workouts help to build global self-esteem, which is an important domain of psychological well-being [156]. Still, if not accompanied by diet, the recommended level of PA may be insufficient for clinically significant weight reduction and maintenance.

Guidelines on weight management differ among countries. A modest weight loss is expected after 150 min of moderate-intensity exercise weekly. Better results may be achieved when training 225–420 min per week without dietary restrictions [157]. However, a recent trial failed to justify the efficiency of long-duration exercise; weight loss did not differ among three groups of women who trained for 72, 135, and 191 min weekly [158]. In contrast, other studies have observed a dose–response relationship between obesity and PA [159].

Individual biological characteristics may account for the variance in response to physical exercise. Twin and adoption studies revealed a strong impact of genetics on BMI [160]. Innate compensatory mechanisms may return BMI to the pre-treatment level after interventions. Obesity can be programmed during in utero development of the fetus; gestational malnutrition is linked with adiposity later in life. After birth, overeating can result from a disruption in the signaling of insulin, leptin, and ghrelin during neurodevelopment [161]. Lifestyle interventions may not be effective if obesity is programmed during gestation and infancy.

The effectiveness of health promotion packages varies across ethnicities. For example, Caucasians and Asians benefit more from lifestyle weight-loss interventions correcting HbA1c levels in people with type 2 diabetes mellitus than Hispanics and Blacks/Africans [162].African American adolescents are more prone to obesity and insulin resistance. The prevalence of risk factors for obesity can also differ among regions and racial/ethnic groups [163]. Policymakers highlight that an effective weight management intervention should cover ethnicity-inclusive risk stratification and treatment strategies [164]. The proposed meta-analysis promotes interventions that take into account biological responses to PA and personal attitudes towards physical exercise shaped by socio-economic status and cultural norms.

## 6. Conclusions

The prevalence of obesity is uneven across geographic regions. The epidemic has reached a plateau in Western countries and shifted to Asia and sub-Saharan Africa. Lifestyle modifications in certain regions may account for the shift in the epidemic landscape. Over 80% of children display sedentary behavior because they treat an active lifestyle as an option rather than a necessity.Apart from environmental factors, individual constructs may also influence the impact of PA on weight gain. A personalized approach to disease management necessitates the analysis of endogenous and exogenous risks in combination.Recent systematic reviews and meta-analyses have limited generalizability, since they mostly covered a single racial/ethnic group or region/country. Another common limitation is that the studies did not consider race/ethnicity and geography as possible confounders. The proposed meta-analysis will outline trends in PA and BMI of children of different races/ethnicities and geographic regions.

## 7. Strengths and Limitations

The PRISMA-P checklist was used to prepare the protocol, and the study methodology was registered with the PROSPERO international database for systematic reviews.The research will synthesize data on PA and BMI in the pediatric population in six geographic regions and for four ethnic/racial groups.Subgroup analysis will be performed to eliminate the effect of study settings on the pooled analysis.The review has been piloted, the results of which indicated evident disparities in PA and BMI across regions and ethnic groups.

## Figures and Tables

**Figure 1 healthcare-12-01830-f001:**
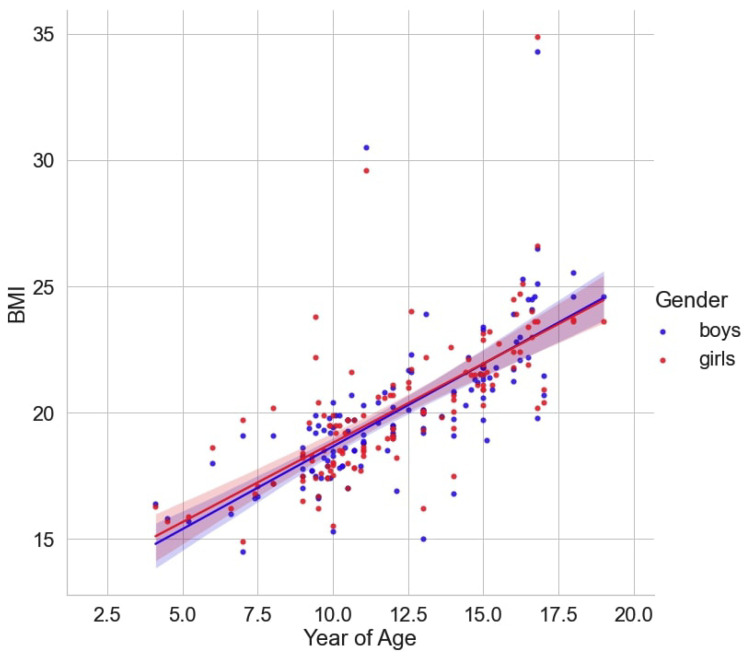
Regression plot between BMI and age.

**Figure 2 healthcare-12-01830-f002:**
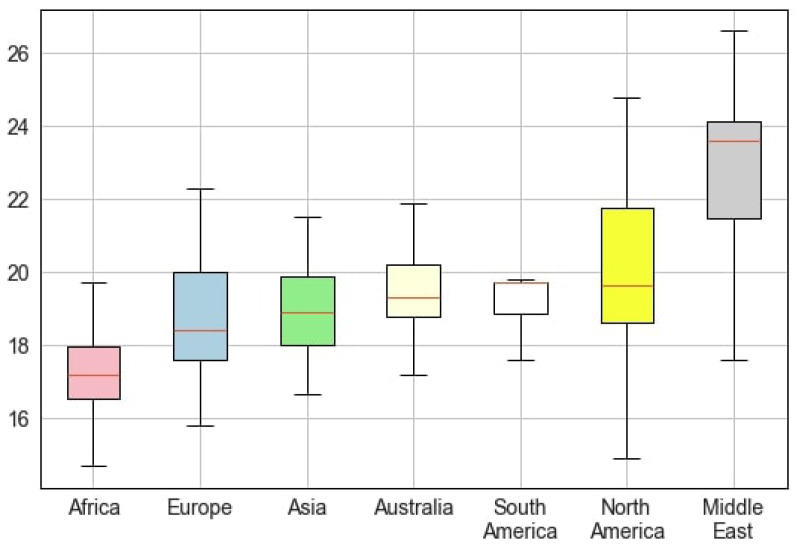
BMI range for both sexes in different regions.

**Figure 3 healthcare-12-01830-f003:**
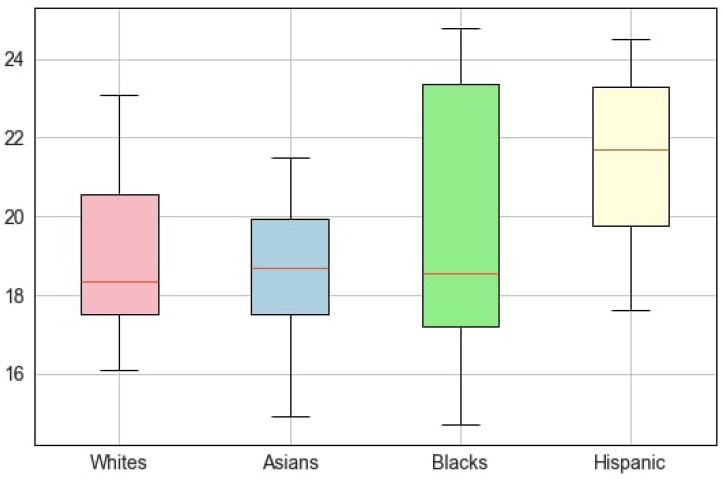
BMI range for both sexes among ethnicities.

**Figure 4 healthcare-12-01830-f004:**
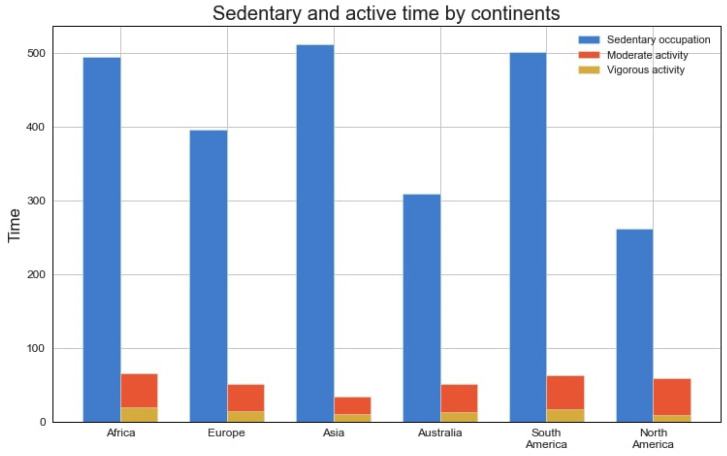
Sedentary and active time by continent (min/day).

**Figure 5 healthcare-12-01830-f005:**
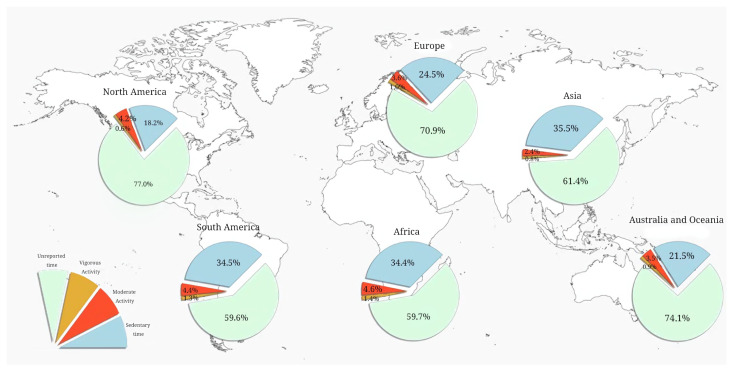
Sedentary and active time by continent (%).

**Figure 6 healthcare-12-01830-f006:**
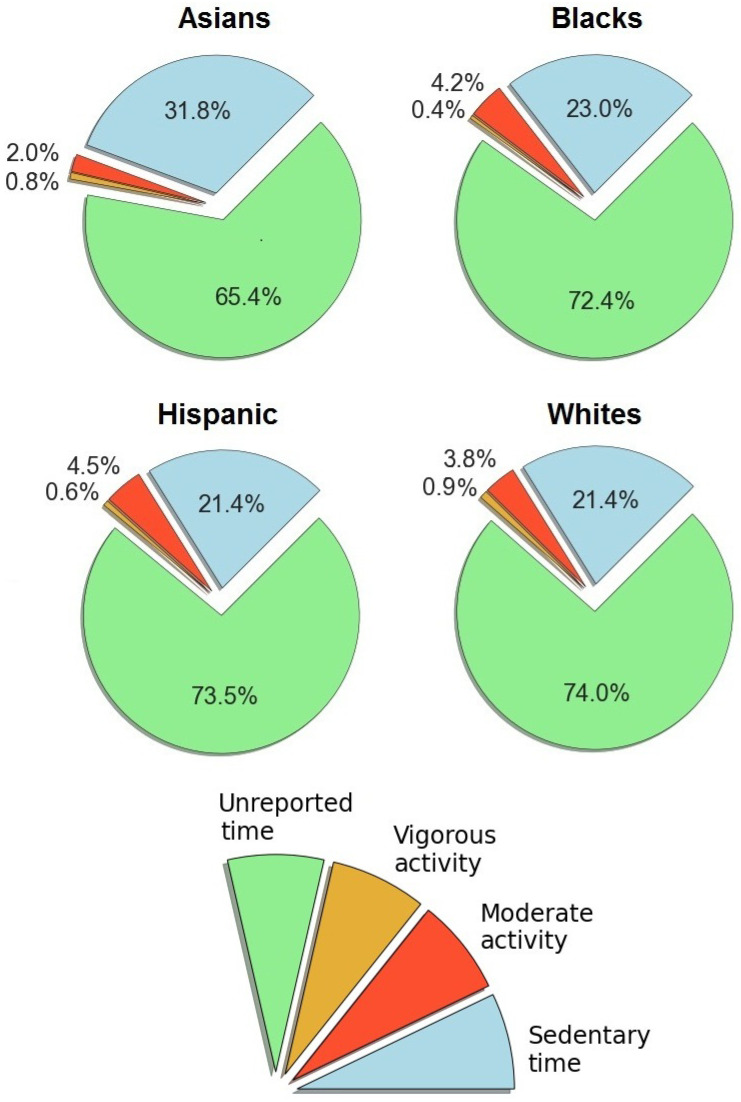
Sedentary and active time by ethnicity (%).

**Figure 7 healthcare-12-01830-f007:**
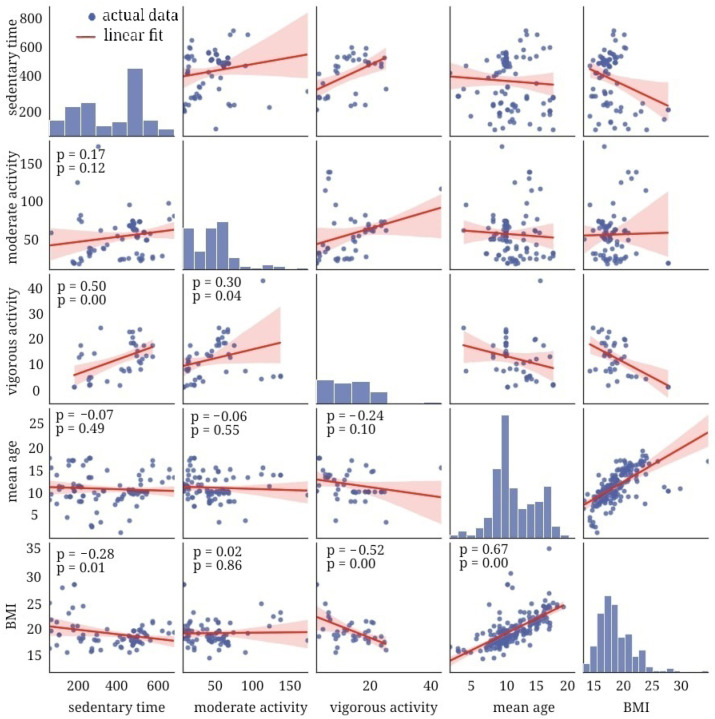
Correlation of BMI with other variables.

**Figure 8 healthcare-12-01830-f008:**
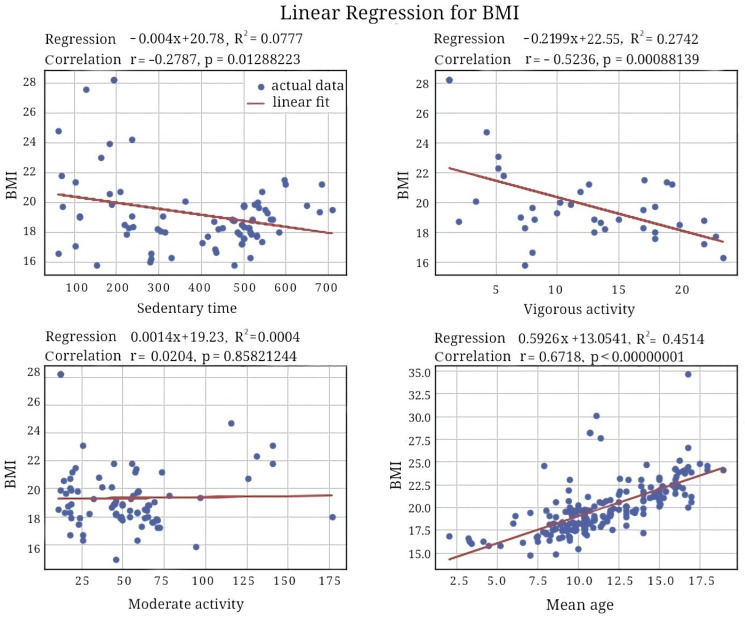
Association of BMI with sedentary behavior and active time.

**Table 1 healthcare-12-01830-t001:** Sedentary and active time by ethnicity.

		Asians	Blacks	Hispanics	Whites	*p*-Value
Sedentary time	Total	458.35 ± 116.67	331.52 ± 160.13	308.03 ± 137.57	307.72 ± 170.83	**<0.05**
	Girls	464.52 ± 160.74	364.27 ± 194.64	204.48 ± 100.34	339.26 ± 138.85	**<0.05**
	Boys	452.31 ± 140.52	396.36 ± 211.07	219.21 ± 99.68	360.42 ± 121.50	**<0.05**
Moderate activity	Total	29.42 ± 15.89	60.64 ± 27.92	65.10 ± 37.48	54.12 ± 35.61	**<0.05**
	Girls	20.43 ± 10.05	80.05 ± 73.36	61.01 ± 12.56	44.07 ± 19.86	**<0.05**
	Boys	26.14 ± 11.36	92.42 ± 59.79	97.30 ± 45.68	57.63 ± 27.58	**<0.05**
Vigorous activity	Total	10.80 ± 3.11	5.93 ± 4.49	9.04 ± 7.22	13.06 ± 13.47	**<0.05**
	Girls	8.22 ± 2.83	11.79 ± 7.60	11.57 ± 9.15	13.24 ± 10.58	**<0.05**
	Boys	12.00 ± 3.42	15.69 ± 7.31	16.54 ± 12.37	18.13 ± 13.34	**<0.05**
Significant differences between cohorts are marked in bold.						

## Data Availability

Data are contained within the article and Appendix A.

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
