# Peer review of "Race, Ethnicity, and Geography as Determinants of Excessive Weight and Low Physical Activity in Pediatric Population: Protocol for Systematic Review and Meta-Analysis"

_healthcare, 2024, doi:10.3390/healthcare12181830_

Round 1
Reviewer 1 Report
Comments and Suggestions for Authors
Even if it's somewhat obvious, I think it's important to justify why the search should involve articles published from 2000 onwards. Likewise, I recommend extending the limit to at least July 2004.
Regarding the eligibility criteria, I recommend including only full articles published in peer-reviewed journals. I believe this will make the quality of the work more robust.
I recommend searching other databases, such as Scielo and Lilacs, as their regional status (Latin America and the Caribbean) can add substantially to the review.
Regarding the formulation of the meta-analysis, I recommend a great deal of caution when building the model, which will mostly involve cross-sectional studies. As well as taking care with the precedence between exposure and outcome variables, I recommend that the authors also justify in the protocol: (I) How will the results of univariate analyses be compared with the results of multivariate analyses? (II) How will the data from multivariate analyses that used different adjustment variables and different regression models be compared? (III) How will articles with different methodological qualities (or risk of bias) be compared?
Reviewer 2 Report
Comments and Suggestions for Authors
The introduction provides an effective contextualization of the global prevalence of childhood obesity and its increasing trend. The introduction could be enhanced by a more explicit statement of the gaps in current research that this study aims to address and an elaboration on the conceptual and practical significance of the study's aims.
It would be beneficial for the authors to provide a table that outlines the characteristics of the selected articles.
In the Method section, the authors employed a random effects model and a ridge regression model. The rationale behind the selection of these particular methods should be provided.
A pilot study is a notable strength of this study; however, the authors should provide a more explicit explanation of how the pilot results influenced the final study design or adjustments made in the Methods section.
The authors must address the statistical significances of 'age- and sex-related differences in growth trajectories of BMI' and 'regional disparities in BMI'.
The figures, 5, 7, and 8, are difficult to read.
In the discussion section, the authors must provide a more explicit link between the study's findings and their public health implications, as well as potential interventions.
Reviewer 3 Report
Comments and Suggestions for Authors
Dear Authors,
thank you for this exciting manuscript! Here there are some consideration and suggestions to empower it:
Overall considerations
The proposed study demonstrates a strong methodological foundation. The adherence to PRISMA-P guidelines and PROSPERO registration ensures transparency and reproducibility. The inclusion of a diverse population and the planned use of advanced statistical methods are commendable and will likely contribute to the study's impact. The protocol also appropriately addresses ethical considerations.
Suggestions to improve the manuscript
To further strengthen the study, consider enhancing the introduction section by including a discussion around lines 39 and 40 on the effects and benefits of resistance training in the pediatric obese population. To support this, you may reference the following articles:
1. Fanelli E, Abate Daga F, Pappaccogli M, Eula E, Astarita A, Mingrone G, Fasano C, Magnino C, Schiavone D, Rabbone I, Gollin M, Rabbia F, Veglio F. "A structured physical activity program in an adolescent population with overweight or obesity: a prospective interventional study." *Appl Physiol Nutr Metab*. 2022 Mar;47(3):253-260. doi: 10.1139/apnm-2021-0092. Epub 2021 Oct 27. PMID: 34706211.
2. Schranz, G., Tomkinson, N., Parletta, J., Petkov, and T. Olds. "Can resistance training change the strength, body composition and self-concept of overweight and obese adolescent males? A randomised controlled trial." *Br. J. Sports Med.*, vol. 48, no. 20, pp. 1482–1488, 201.
Additionally, consider including factors such as socioeconomic status, access to healthcare, dietary habits, and cultural differences as potential confounders in the analysis or explain the possible bias in data interpretation given by they exclusions
Finally, this study presents a promising research design with the potential to make a valuable contribution to the field. With the incorporation of the suggested improvements, the study can be significantly strengthened.
Thank you again for this interesting paper
Round 2
Reviewer 2 Report
Comments and Suggestions for Authors
The authors have successfully addressed comments, and the manuscript has been improved sufficiently for publication. Thank the authors for improving the manuscript.